# Calculation of coseismic displacement from Lidar data in the 2016 Kumamoto, Japan, earthquake

Luis Moya[1], Fumio Yamazaki[1], Wen Liu[1], Tatsuro Chiba[2]

[1]Department of Urban Environment Systems, Chiba University, Chiba 263-8522, Japan
[2]Research and Development Institute, Asian Air Survey Co., Ltd., Kawasaki 215-0044, Japan

*Correspondence to*: Luis Moya (lmoyah@uni.pe)

**Abstract.** The spatial distribution of the coseismic displacements that occurred along the Futagawa fault during the 2016 Kumamoto earthquake of $M_w$ 7.0 was estimated using airborne light detection and ranging (Lidar) data. In this study, a pair

of digital surface models (DSMs) obtained from the high-density Lidar data before and after the mainshock on April 16, 2016, was used. A window matching search approach based on the correlation coefficient between the two DSMs was used to estimate the geodetic displacement in the near-field region. The results showed good agreements with the geodetic displacements calculated from strong-motion acceleration records and coincided with the fault line surveyed by the Geological Survey of Japan.

**Keywords:** Coseismic displacement, Lidar, the 2016 Kumamoto earthquake, acceleration record

## 1 Introduction

On April 14, 2016, an $M_w$ 6.2 earthquake struck Kumamoto Prefecture, Japan, at 21:26 JST. The epicenter was located at the end of the Hinagu fault at a shallow depth. After approximately 28 h (at 01:25 on April 16, 2016), another earthquake of $M_w$ 7.0 struck the Futagawa fault, which is near the Hinagu fault. The first event was designated as the foreshock and the second

one as the mainshock. Both the events occurred in the town of Mashiki (having a population of approximately 33,000), which is located to the east of Kumamoto City (having a population of approximately 735,000). Many aftershocks followed these events, and as of September 6, four months after the foreshock, the total number of aftershocks (larger than Mw 3.5) is 272. This number is the largest among the recent inland (crustal) earthquakes in Japan (Japan Meteorological Agency, 2016). This Kumamoto earthquake sequence triggered secondary effects such as landslides and liquefaction, and caused extensive

damage to lifeline systems, buildings, bridges and transportation structures. A total of 8,550 buildings, mostly in Kumamoto Prefecture, were seriously damaged or collapsed, and 50 human lives were lost, mostly because of landslides or the collapse of buildings (Cabinet Office of Japan, 2016).

Soon after the occurrence of the foreshock, various satellites and airborne remote sensing technologies were employed to monitor crustal movements and various damages (Yamazaki and Liu, 2016). The Japan Aerospace Exploration Agency

(JAXA) carried out extensive monitoring of the source area using the PALSAR-2 sensor on board ALOS-2 satellite.

Interferometric synthetic aperture radar (InSAR) analysis using a pair of data obtained from PALSAR-2 before (pre-event data) and after (post-event data) the mainshock showed the line-of-sight displacements to the satellite direction (Geospatial Information Authority of Japan, 2016). Using the pre-event data (November 30, 2015, March 07, 2016) and the co-event data (March 07, 2016, April 18, 2016) from PALSAR-2, the authors of this paper calculated the spatial coherence values

(International Charter, 2016), which could highlight the extensive landslides and severe damages to buildings along the Futagawa fault line.

After the Kumamoto earthquake, government agencies and aerial survey companies in Japan conducted several aerial surveying flights such as high-resolution vertical and oblique aerial photography, and airborne light detection and ranging (Lidar) surveys (Asia Air Survey Co., Ltd., 2016; Geospatial Information Authority of Japan, 2016).

The airborne Lidar technology is an integrated system consisting of a Global Navigation Satellite System (GNSS), an Inertial Navigation System (INS) and a laser scanner, which sends pulses of laser light towards the ground and records the return time for calculating the distance between the sensor and the ground surface (Lillesand et al., 2004). Lidar has many applications in earthquake engineering, such as landslide detection (Jaboyedoff et al., 2012), and extraction of building features (Vu et al., 2003; Vu et al., 2009). Lidar data have been used in estimating ground displacement as well. Muller and

Harding (2007) used the elevation of uplifted marine terraces mapped in the Lidar data to estimate the source parameter of the A.D. 900 Seattle fault earthquake. Sahakian et al. (2016) used Lidar data, in combination with other technologies such as seismic reflection, to identify a previously unmapped right-lateral strike-slip fault located in the Salton sea, California, U.S. They used the Lidar data to constrain the onshore deformation.

Usually, only post-event Lidar data is available; thus, the coseismic displacement detection is limited to the identification of

distortions of line features such as roads. Li et al. (2016) detected an offset of car tracks produced during the 2014 Mw 6.9 Yutian earthquake, Tibetan Plateau, by visual inspection. Chen et al. (2015) extracted two topographic profiles from Lidar data collected after the 1999 Mw 7.1 Hector Mine earthquake, California. The profiles were parallel to the fault-line and located on either side of the fault in order to estimate the slip during the earthquake. There are few cases in which Lidar data both before and after an earthquake were available. The first case was in the 2010 Mw 7.2 El Mayor-Cucapah earthquake.

Oskin et al. (2012) performed a simple difference of elevation to estimate the surface rupture; however, they did not consider the horizontal displacement. Two more earthquake events: the 2008 Mw 6.9 Iwate-Miyagi earthquake and the 2011 Mw 7.1 Fukushima-Hamadori earthquake were monitored by Lidar data acquired before and after the event. Then Nissen et al. (2014) estimated the 3D displacement using the Iterative Closest Point (ICP) algorithm (Nissen et al. 2012). Their results showed a coherent displacement but with high level of noise in the horizontal component.

Cross-correlation technique has been used successfully to monitor movements. Duffy and Hughes-Clarke (2005) applied cross-correlation to monitor the movements of sea–floor dunes using bathymetry data. Liu et al. (2011) extracted the shifts of vehicles between the panchromatic and multispectral QuickBird images, which were taken with a time lag of approximately 0.2 seconds, and then they estimated the vehicles' velocity. Liu and Yamazaki (2013) calculated the crustal displacement during the 2011 Mw 9.0 Tohoku earthquake by estimating the shift of undamaged buildings using the cross–correlation

coefficient between the TerraSAR–X intensity images taken before and after the earthquake. Borsa and Minster (2012) evaluate the potential use of cross-correlation using Lidar data by applying a synthetic slip to the Lidar data of the southern San Andreas fault and then their result could recover the synthetic slip. Duffy et al. (2013) also used a pair of Lidar data taken before and after the 2010 Mw 7.1 Darfield, New Zealand earthquake to calculate the horizontal coseismic displacement.

Measurements of the coseismic displacement in the near field is of great importance because it can be used to locate the source and to understand the rupture process. Wang et al. (2013) inverted the coseismic displacement calculated from GNSS and strong-motion stations to modulate the earthquake source of the 2011 Mw 9.0 Tohoku earthquake. Earthquake source inversion methods have become important in the last years because of its potential for forecasting tsunamis (Melgar and Bock, 2013). The GNSS devices calculate positions and nowadays it is used for continuous monitoring of the earth crust.

Strong-motion devices record acceleration or velocity, and in most of the cases, a baseline correction is required before estimating the correct displacement time history because the baseline is shifted as a result of several factors such as ground rotation and rocking movements of the instrument. The displacement time history can be calculated precisely if the six components, three translational and three rotational, are recorded (Graizer, 2010). However, the displacement time history is often estimated by a double integration of only the translational components with respect to time. Up to now the source of

errors and the rotation components cannot be quantified and only empirical methods have been proposed in the past to reduce the effect of the baseline shift and retrieve a reliable displacement time history. One of the first method was proposed by Iwan et al. (1985), in which a bilinear function is used to estimate the velocity trend caused by the baseline errors. Several modifications of this approach have been proposed. Wu and Wu (2007) defined the bilinear function in an iterative process in a way that the displacement time history best fits a ramp function. Later, Wang et al. (2011) also proposed an iterative

procedure; but they used a step function to constrain the displacement time history. Moya et al. (2016) used a pair of strong-motion records that were closely located and perform a simultaneous correction of both records.

Although there have been a great improvement and deployment of GNSS and strong-motion networks, even the densest network, either GNSS or strong-motion, has a low spatial resolution. For instance, the nationwide GNSS network of Japan

25

has one station in an about 20-km interval. Thus, for an earthquake of moderate magnitude, where the coseismic displacement is concentrated in a narrow area, it is difficult to depict the spatial pattern of coseismic displacement. SAR satellite images offer a better spatial resolution, but it requires a pair of images with the same viewing condition to calculate the coseismic displacement to the line-of-sight (LOS) of radar. More pairs of SAR images from different views, which are not very realistic, are required to obtain 2.5 D or 3D coseismic displacement.

30

Another use of coseismic displacement comes up when the effects of an earthquake in the near field are estimated using remote sensing techniques. It is necessary to consider the permanent displacement if an automatic change detection is applied to extract collapsed buildings or quantify the mass movement in landslides.

This paper estimates the coseismic displacement due to the mainshock of the Kumamoto earthquake using the Digital Surface Models (DSMs) obtained from airborne Lidar flights (Asia Air Survey Co., Ltd., 2016). In this case study, a pair of

DSMs, one soon after the foreshock (on April 15, 15:00 - 17:00 UTC+09:00) and another after the mainshock (April 23, 10:00 - 12:00 UTC+09:00), corresponding to the town of Mashiki, which includes the causative Futagawa fault, were used. The obtained results are compared with the permanent ground displacements estimated from fields surveyed data and using the acceleration records obtained from KiK-net, K-NET, the strong-motion seismograph network of Kumamoto Prefecture, and a temporary observation system (Hata et al., 2016).

## 2 Study area and data description

On April 15, 2016, one day after the big foreshock, a Lidar DSM was acquired to record the surface rupture and various effects of the earthquake, such as buildings damaged and landslides (Asia Air Survey Co., Ltd., 2016). The survey generated a DSM of average point density 1.5–2 points/m$^2$. Furthermore, because of an unexpected mainshock of $M_w$ 7.0 on April 16, a second mission was set up on April 23 to acquire Lidar data. The second survey was able to generate a DSM of average point density 3–4 points/m$^2$. After the rasterization of the raw point clouds, the DSMs have a data spacing of 50 cm and are registered to the Japan Plane Rectangular Coordinate System. This data set is one of the few cases in which pre- and post-event DSMs are acquired by the same pilot using the same airplane and instrument. For the sake of brevity, we will call the DSMs acquired on April 15 and April 23 as the pre-event DSM and the post-event DSM, respectively.

Figure 1 illustrates the extension of these two DSMs in which the pre-event DSM extends to a bigger area than the post-event DSM does. The common area between both the DSMs covers most parts of Mashiki town and a few parts of Kashima town, Mifune town, and Nishihara village with elevations ranging from 1 m to 500 m (Figure 2). The entire common area is composed of residential buildings, agricultural fields, forests, and a part of the Futagawa fault that caused the mainshock of the Kumamoto earthquake.

The Kumamoto earthquake occurred in an area that is sufficiently equipped with several GNSS instruments that belong to GEONET (Sagiya, 2004) and strong-motion instruments that belong to KiK-net, K-NET (Aoi et al., 2004), the strong-motion seismograph network of Kumamoto Prefecture, and a temporal network deployed by Hata et al. (2016). Figure 1 indicates the location of all the stations within and near the study area. GEONET consists of approximately 1,300 GNSS control stations that cover the entire territory of Japan with an average interval of 20 km. K-NET consists of more than 1,000 strong-motion accelerometers installed on the ground surface at every 20 km covering Japan. KiK-net consists of approximately 700 stations and each station has a pair of accelerometers installed on the ground surface and in a borehole in bedrock. The strong-motion seismograph network of Kumamoto Prefecture consists of strong-motion accelerometers installed at the municipality building sites.

The evidence of coseismic displacements has been observed in the form of surface ruptures in agriculture fields, river channels, and roads along the Futagawa fault line during the Kumamoto earthquake (Figure 3). The surface ruptures were caused by the opposite displacements (right-lateral strike slips) between both the sides of the fault. A comparison of the pre-event DSM with the post-event DSM gives a clearer evidence of the coseismic displacements. Figure 4 shows an overlap of

the two DSMs where the pre- and post-event DSMs are represented by cyan and red colors, respectively. The gray-colored pixels represent the locations that have the same elevation in both the pre- and post-event DSMs, whereas the cyan-colored pixels represent the locations that have a higher elevation in the pre-event DSM and the red-colored pixels represent the locations that have a higher elevation in the post-event DSM. Therefore, the colors around the sides of the houses depicted in Figure 4b show that the coseismic displacement occurred to the northeast direction.

## 3 Methodology

To calculate the horizontal component of the coseismic displacement distribution in space, we introduced a maximum correlation search algorithm using a moving window of the post-event DSM within a corresponding larger area of the pre-event DSM. The method is developed based on the fact that both the pre- and post-event DSMs cover the same objects, such as non-damaged buildings. This fact can be used most efficiently for calculating the spatial cross-correlation between the DSMs. At any location, the pixel shift necessary to match the pre-event DSM with the post-event DSM is assumed to be the coseismic displacement at the location. However, the coseismic displacement is variable in space and has to be calculated using sub-areas (windows). Figure 5 shows a scheme of the coseismic displacement search method. First, we consider a square sub-area of the post-event DSM and a bigger sub-area of the pre-event DSM with their centers located at the same coordinate (Figure 5a). Then, we reduce the pixel size using a cubic convolution method (Figure 5b). The post-event window is moved across the pre-event window, and the cross-correlation coefficient is calculated for the moving area (Figure 5c). The location of the pixel that has the largest correlation value is considered as the coseismic displacement for that window. The horizontal component of the coseismic displacement was applied to the post-event DSM to cancel it, and then the vertical displacement between the two DSMs was calculated. It is worth mentioning that the cross-correlation was chosen among other candidates, such as a least-square difference or convolution, mainly because the peak value was located in a narrower area.

It is not necessary to calculate the correlation for all the locations because it requires unnecessary computational efforts. A better procedure is to move the post-event window along the direction in which the cross-correlation is increasing faster until the peak is reached. This approach, well known as the steepest method, was applied to calculate the coseismic displacement for all the study areas. Thus, in this approach, only the size of the post-event window has to be defined and the rest are done automatically. However, selecting the size of the post-event window is crucial because the window should be large enough to include several distinct objects. For instance, if a post-event window of 1.5 m x 1.5 m (3 x 3 pixels) is chosen, the peak value of cross-correlation might not be obtained when the window is located in the middle of a flat building roof or a big bare land. Therefore, it is recommended to define a window that includes some buildings or different topography. However, there exists a trade-off between the  size of the window and resolution because the resolution of the spatial variation of the coseismic displacement decreases with the increase in the size of the window.

The code for implementing the method was written in Python, an open-source programming language, in order to use the large collection of scientific open-source modules. Numpy, a numerical array-programming module, was used to calculate the cross-correlation. Open-CV (Open Source Computer Vision Library) was used to reduce the resolution of pixels using the cubic convolution method. GDAL (Geospatial Data Abstraction Library) was used to georeferenciate all the inputs and
outputs.

## 4 Result of analysis

Using the methodology explained above, we estimated the coseismic displacements in the common area between the pre- and post-event DSMs, which is approximately 80 km$^2$. The pixel resolution was increased from 50 cm to 10 cm by using the cubic convolution method, where a bicubic function is fitted using 4x4 pixels neighborhood and used to estimate the
intermediate values. The subpixel size was decided based on the computational effort that is required to detect the peak value of the correlation coefficient. The size of the window was decided based on the area required to cover several objects in the DSMs. Figure 6 compares the east-west coseismic displacement obtained using a window of size 201 x 201 pixels with that obtained using a window of 101 x 101 pixels. The results obtained using a window of size 101 x 101 pixels indicate increased noise level in the areas of large agricultural fields because the peak of the correlation coefficient cannot be
identified clearly. On the other hand, a window of size 201 x 201 pixels covers an area large enough to reduce the noise substantially. Thus, a window of size 201 x 201 pixels (100.5 m x 100.5 m) was selected for the overall study area. Another issue is to evaluate the magnitude of the maximum correlation coefficient, which is used to identify the coseismic displacement. Figure 7 illustrates a histogram of the maximum correlation coefficients detected for each window. The left vertical axis shows the number of observations per 0.01 intervals of the correlation coefficient and the right vertical axis is
for the cumulative frequency. The figure indicates that most of the results produced a large correlation coefficient and a closer look revealed that the areas with a correlation coefficient less than 0.6 showed the results not consistent with the surrounded areas; however, only 14 cases out of 9,195 windows produced a correlation coefficient less than 0.6.

Figure 8 shows the east-west and north-south components of the coseismic displacement with a certain level of noise, which is mainly because some objects are not exactly the same after the earthquake. Several buildings collapsed and landslides
occurred as a result of the mainshock. Besides, the post-event DSM contains certain objects that were not present in the pre- event DSM, such as the vehicles and tents used as shelters. However, the general trend of spatial variation of the coseismic displacement could be depicted adequately. The spatial distribution of the three-dimensional (3D) coseismic displacement is shown in Figure 9. The black arrows indicate the 2D horizontal component and the color shading indicates the vertical displacement. In order to show only the vertical coseismic displacement and remove the effect of the collapsed buildings and
landslides, a median filter with a window of the same size (201 x 201 pixels) as the one used for the matching method was applied. Thus, the resolution of the horizontal displacement is the same as that of the vertical displacement. Although the output provided coseismic displacements in a 100.5-m grid, the black arrows show the displacements only at every 500 m in

order to visualize the orientation of the coseismic displacement efficiently. The change of direction of the coseismic displacements in both the horizontal and vertical planes delineates the Futagawa fault line, which is consistent with the surveyed active faults in Japan and the results of the field investigations conducted by the Geological Survey of Japan (2016). The observed coseismic displacement shows eastward movements of up to 2.0 m in the northern area and 1.2 m in the southern area of the fault line. The legend of the vertical displacement shows a vertical displacement of up to -3 m; however, this value corresponds to a narrow area where a large landslide occurred and the median filter could not remove it completely.

A closer look at the general trend shows that a subsidence of up to 2 m occurred in the northern area and an uplift of up to 0.7 m in the southern area. Our results are consistent with the coseismic displacement estimated by using SAR interferometry using ALOS-2 PALSAR-2 imagery (Geospatial Information Authority of Japan, 2016). Figure 10 shows the coseismic displacement profiles corresponding to the eight dashed lines that are drawn uniformly along the Futagawa fault (see the locations in Figure 9). The changes in the direction of the displacement for all the components are located almost at the same point, the surveyed Futagawa fault line. However, the change of sign occurs gradually because the applied window contained points from the both sides of the fault line and consequently produced small coseismic displacements. The main deformation was caused by the slip at the main Futagawa fault line; however, the profiles GH and IJ show smaller slips caused by the secondary Futagawa fault line.

## 5 Validation of results

The coseismic displacements obtained from the Lidar DSMs were compared with that obtained from the other sources of information. Currently, the GNSS technology is used to monitor crustal deformation within a centimeter-level accuracy. Unfortunately, there is no GEONET station in this study area (Figure 1). However, several strong-motion instruments whose results can be used to compare with that of the Lidar data are available. The distribution of six strong-motion stations located within the study area is shown in Figure 9. One station, with code KMMH16, belongs to KiK-net and two stations belong to the strong-motion seismograph network of the prefecture: one located at the Mashiki town office (MTO as referred by Hata et al., 2016) and the other at the Nishihara village office (hereafter, NVO). Three stations, TMP1, TMP2, and TMP3, belong to a temporary network deployed by Hata et al. (2016) with the objective of monitoring the aftershocks following the event on April 14. The mainshock of Mw 7.0 occurred after the deployment of the temporary network, and the acceleration records from the stations in this network were acquired successfully. Furthermore, a K-NET Kumamoto station, with code KMM006, is located 1 km from the closest point of the study area. Digital acceleration records obtained from these seven stations could be used to estimate the coseismic displacement caused by the mainshock.

The method proposed by Wang et al. (2011) was applied to the acceleration records obtained from the seven strong-motion stations mentioned above. The baseline correction procedure estimates a bilinear function from the uncorrected velocity time

history, which is obtained by integrating the acceleration with respect to time. For instance, Figure 11 shows the baseline correction estimated using the uncorrected velocity obtained from the NVO station. Then the bilinear function is removed from the uncorrected velocity and the displacement is calculated. The coseismic displacement calculated from the Lidar data at the same location of the strong-motion station, shown as a black thick line, is very close to the permanent displacement

observed from the displacement time history. Figure 12 depicts the coseismic displacements at the MTO, KMMH16, and KMM006 stations obtained from the acceleration records and the Lidar data. The figure reveals that the coseismic displacements derived from the DSMs are consistent with those obtained from the strong-motion acceleration records. However, they are not exactly the same because of the fact that the double integration of acceleration is empirical and it can provide only an approximation. In the case of the K-NET Kumamoto station, the results are compared with that obtained

from the closest DSM, which is approximately 1 km away. There were two accelerometers at the KiK-net KMMH16 station, one on the ground surface and the other in a borehole (-252 m below the surface). Although the two permanent displacements were calculated independently, both the results were similar to that obtained from the Lidar data. This fact validates the method proposed by Wang et al. and the accuracy of the results obtained from the Lidar DSMs.

On the contrary, the coseismic displacements obtained from the acceleration records at TMP1, TMP2, and TMP3 were

different from those obtained from the Lidar data (Figure 14). This large discrepancy is because the instruments at TMP1, TMP2, and TMP3 were placed on the ground surface without foundation. Thus, they did not have sufficient confinement to avoid movements relative to the ground, such as rocking or rotation around the vertical axis. Therefore, the displacements obtained from the temporary network could not be estimated using just two linear segments in the uncorrected velocity, which is the method proposed by Wang et al. (2011). These additional distortions can be easily observed in the north-south

component at the three stations.

Another source of information that can be used to compare our results is the report of field surveys performed by the Geological Survey of Japan (GSJ). In Figure 14, red lines indicate the surface ruptures surveyed by the GSJ and the black arrows indicate the direction of displacement together with the amplitude range of the slip. Figure 6a illustrates the surface rupture lines together with our results for the east-west component. Ten profiles, in which the displacements were measured

by the GSJ, were used to calculate the displacements parallel to the fault lines (Figure 15).

# 6 Discussion

Our result could recover the spatial distribution of the three-dimensional (east-west, north-south, and up-down) coseimsic displacement and validated the fault line drawn by the GSJ (Figure 6, 8 and 9). From the evaluation of the parameters used, the results were found to be highly sensitive to the window size. Basically, it is crucial that the windows have to cover

several features, such as buildings, trees and different topography, in order to obtain a clear peak value in the correlation coefficient (Figure 5c). This issue was our main concern in agricultural fields because large areas have uniform elevation. In this study, a constant window size was used; however, if the land use information is available, different window sizes can be

applied. For instance, in urban areas the window size can be smaller than that for agricultural lands. Therefore, one limitation of the method is the required window size because the larger the window size, the lower the spatial resolution of coseismic displacement.

Comparing our result with the InSAR satellite images published by the GSI, our result provides the 3D coseismic displacement; while the InSAR results provide only the displacement to the line-of-sight. But concerning about the area coverage, satellite sensors can cover a larger area than airborne Lidar sensors do.

The slips calculated from our results are very close to that obtained from the field observation for most cases (Figure 14 and 15). It is observed that in the majority of the cases our results are greater than the measured ones. We believe that the main reason for this is that the type of soil is cohesive in this area. Cohesive soils have the ability to exhibit large plastic deformation that depends on the water content and, as can be seen, the area is mostly used for agricultural purposes where the soil has high water content. Thus, the surface rupture measured in the field might not be the total slip. The largest differences between the GSJ survey and the Lidar results are observed in the profiles 'op' and 'qr'.

Lidar data are capable of extracting other types of information. Figure 16 shows two areas: one with collapsed buildings and the other where a landslide occurred. Figure 16 also shows the change in elevations between the DSMs after removing the horizontal coseismic displacement. As can be observed, the large change in elevations implies that a building collapsed or a landslide occurred. Therefore, with proper thresholds, these phenomena can be detected automatically. This issue will be discussed in a future publication.

## 7 Conclusions

The coseismic displacements produced during the mainshock of $M_w$ 7.1 of the 2016 Kumamoto earthquake were estimated using two DSMs acquired by high-resolution Lidar flights before and after the mainshock on April 16. The common area between the DSMs covers approximately 80 km$^2$ including the Mashiki town section of the known Futagawa fault line. The maximum cross-correlation coefficient was used with a window matching technique between the two DSMs to calculate the coseismic displacement. With a window of size 100 m x 100 m, the maximum cross-correlation value reached more than 0.6 for more than 99.8% of the all 100-m grid points. Coseismic horizontal displacements of up to 2 m and subsidence of up to 2 m were observed in the study area. These values are the largest coseismic displacements produced during the Kumamoto earthquake, which were not recorded at any GEONET stations. The results showed good agreement with the permanent displacements calculated from the double integration of the strong-motion accelerations at the seven seismic stations. The results were further compared with the surface ruptures observed by the GSJ, and a reasonable level of agreement was reached in terms of location and slip amplitude along the Futagawa fault.

The detailed information of coseismic displacement is indeed useful to constrain the focal mechanism of the event. Recall that the GSI's preliminary report estimated a slip of about 24 m in the source zone during the 2011 Mw 9.0 Tohoku earthquake from an inversion method using the inland GEONET station records. However, later Sato et al. (2011) observed a

coseismic displacement of 23 m at the ocean bottom and pointed out that this information could better constrain the focal mechanism. Thus, our results, which records higher coseismic displacement than the one recorded from GNSS stations, would improve the source estimation. However this issue is out of the scope of this paper and will be addressed in a future publication.

As mentioned before, there are only few cases in which Lidar data before and after an earthquake are available. The main reason is a high cost of Lidar surveys. However, this technology can be used properly for a specific region of interest, such as along fault lines. For instance, the B4 project (Bevis et al., 2005) collected Lidar data of the southern San Andreas and San Jacinto faults in southern California in order to have a pre-event Lidar data for future earthquakes.

## 8 Data and Resources

Strong-motion data collected from KiK-net and K-NET can be accessed online at http://www.kyoshin.bosai.go.jp/ (last accessed August 2016) and strong-motion data from the strong-motion seismograph network of Kumamoto Prefecture were released via the Japan Meteorological Agency (JMA) at http://www.data.jma.go.jp/svd/eqev/data/kyoshin/jishin/1604160125_kumamoto/index2.html (last accessed August 2016). The temporary observation records in the town of Mashiki were obtained from the works of Hata et al. (2016) at

http://wwwcatfish.dpri.kyoto-u.ac.jp/~kumaq/ (last accessed August 2016). The Numpy library can be accessed at http://www.numpy.org/# (last accessed August 2016), the OpenCV library can be accessed at http://opencv.org/ (last accessed 2016), and the GDAL library can be accessed at http://www.gdal.org/index.html (last accessed August 2016).

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

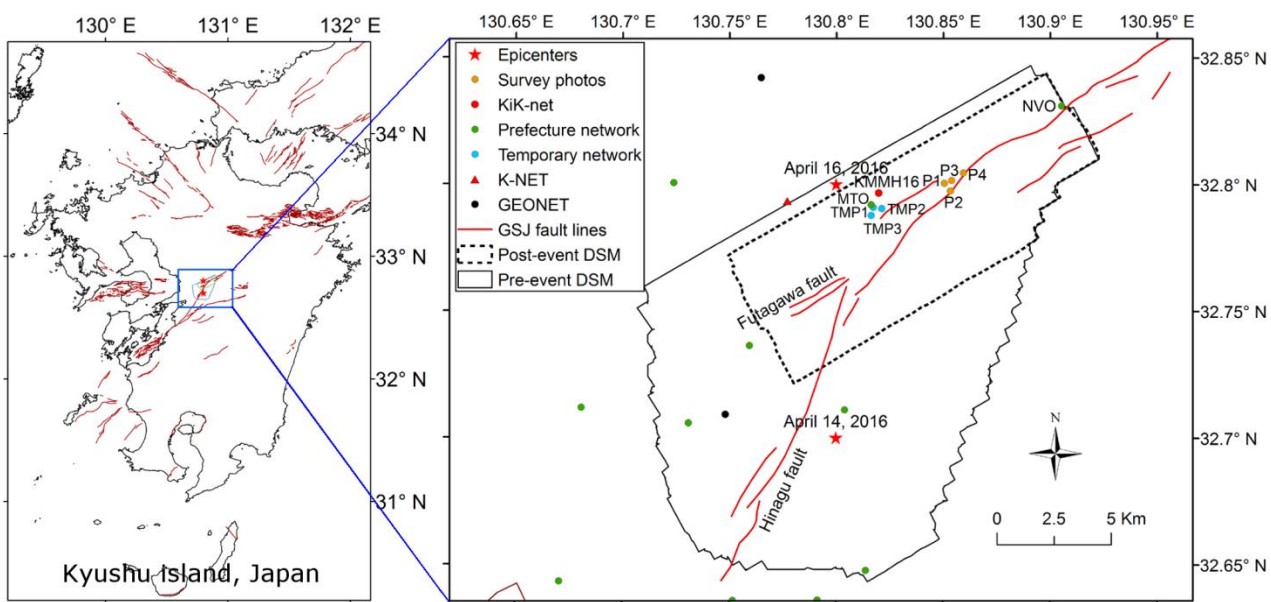

**Figure 1. Map of the near-source area of the 2016 Kumamoto earthquake, showing the areas of the pre-event DSM (black solid polygon) and the post-event DSM (black dashed polygon), the distribution of the GNSS and seismic stations, active fault lines in Japan (red lines), and epicenters (M$_w$ 6.2 April 14, 2016; M$_w$ 7.1 April 16, 2016).**

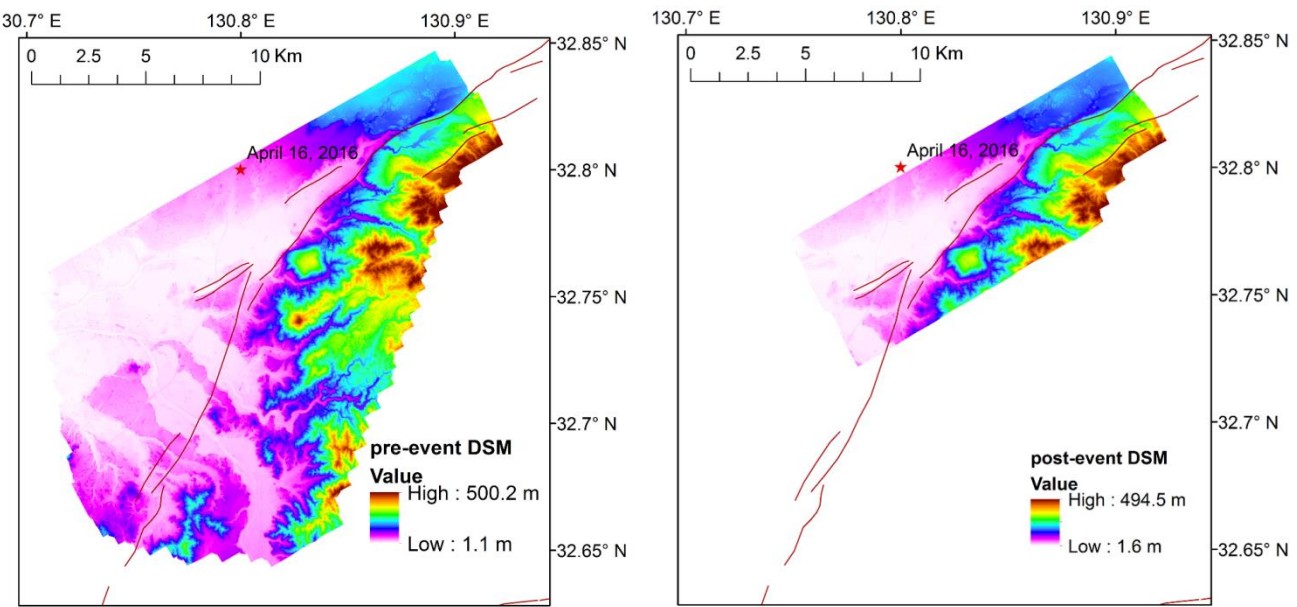

**Figure 2. DSMs acquired by Asia Air Survey Co., Ltd. (2016) on April 15, 2016 (pre-event DSM) and April 23, 2016 (post-event DSM).**

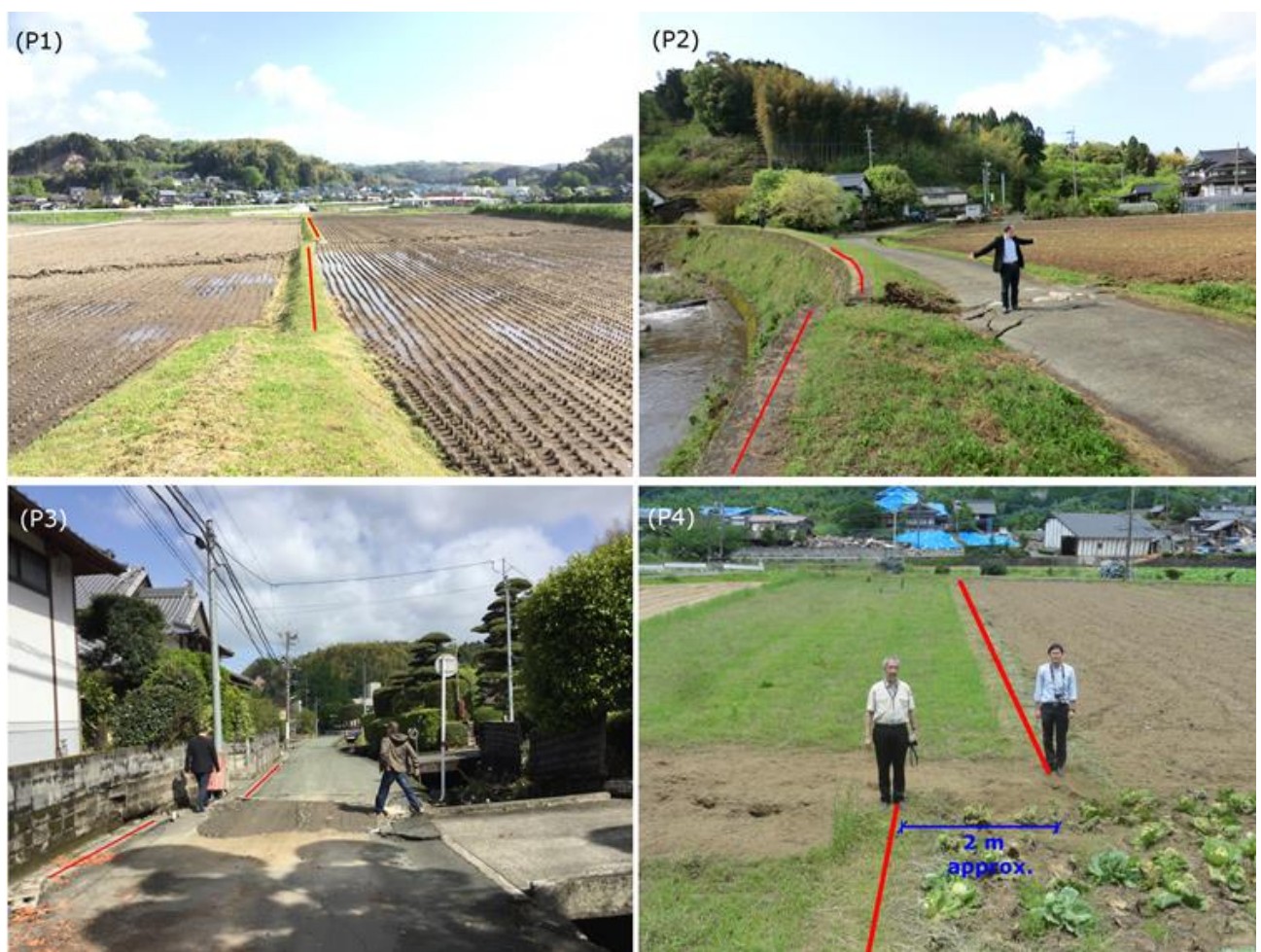

**Figure 3. Examples of surface ruptures caused by the 2016 Kumamoto earthquake. Paddy field (P1), river channel (P2), road crossing in Kamijin and Shimojin districts of the town of Mashiki observed on April 17, 2016 (P3), and crop field in Dozono district of the town of Mashiki observed on June 7, 2016 (P4). The locations of the photographs are shown in Figure 1.**

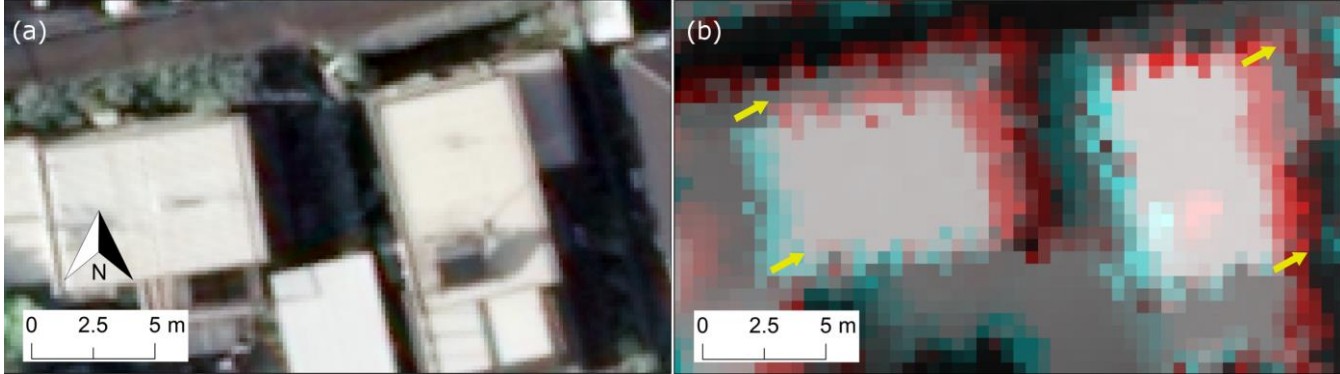

**Figure 4. Example of coseismic displacement extracted from Lidar data: (a) Aerial image of buildings near the Mashiki KiK-net station acquired on April 15, 2016, and (b) color composite of the post-event (red) and pre-event (cyan) DSMs for the same area where the yellow arrows depict the direction and amplitude of the coseismic displacement.**

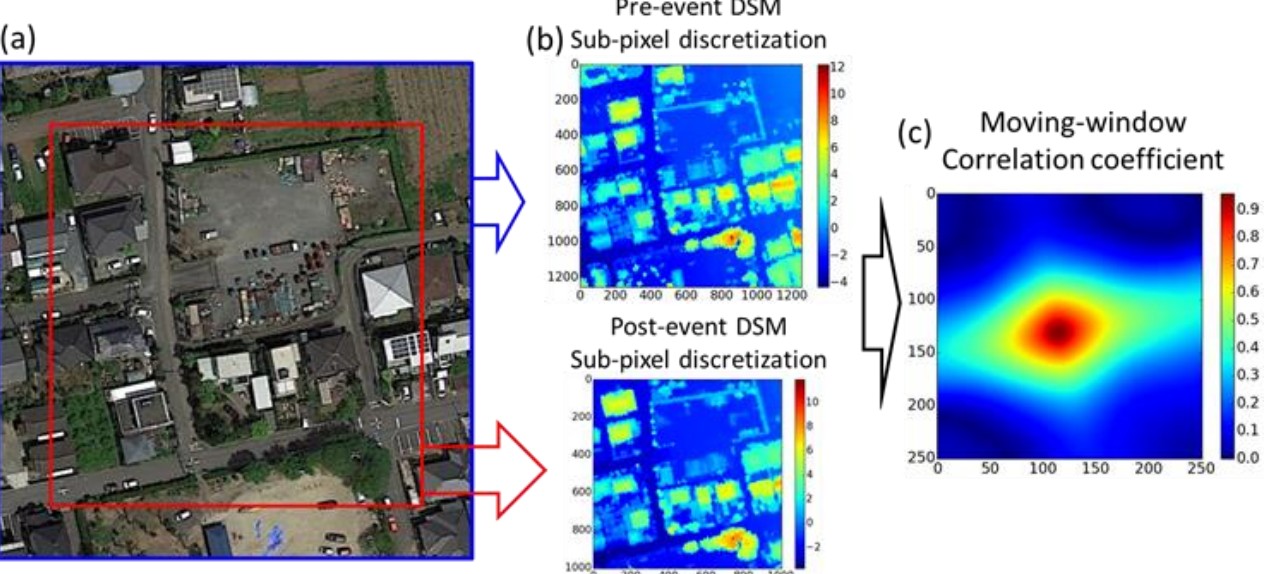

**Figure 5. Schematic image of the maximum correlation search algorithm. Selection of the pre-event DSM (blue) and post-event DSM (red) windows (a), sub-pixel discretization of the DSMs (b), and calculation of correlation coefficient by moving the window of the post-event DSM over the pre-event one (c).**

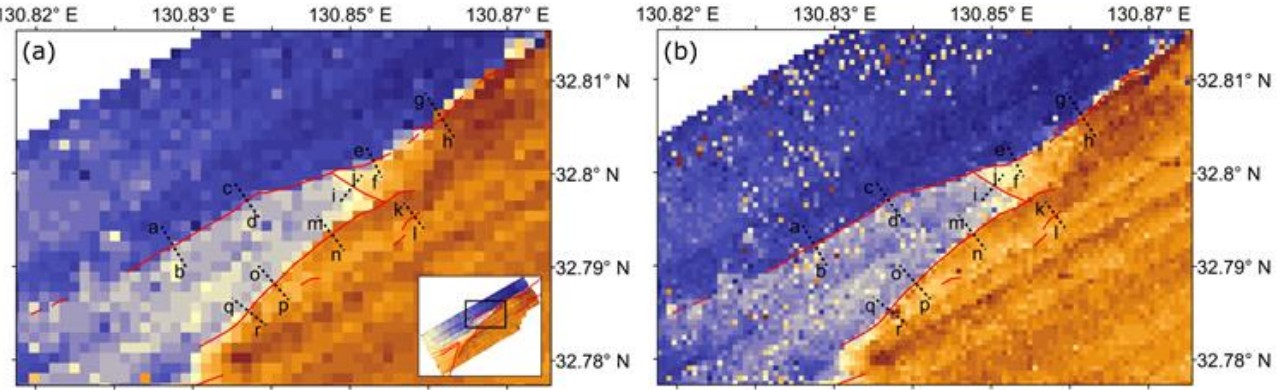

**Figure 6. Illustration of noise generated in the coseismic displacement for a window of size 201 x 201 pixels (a) and 101 x 101 pixels (b). The black square in the inset map shows the area of the main figure.**

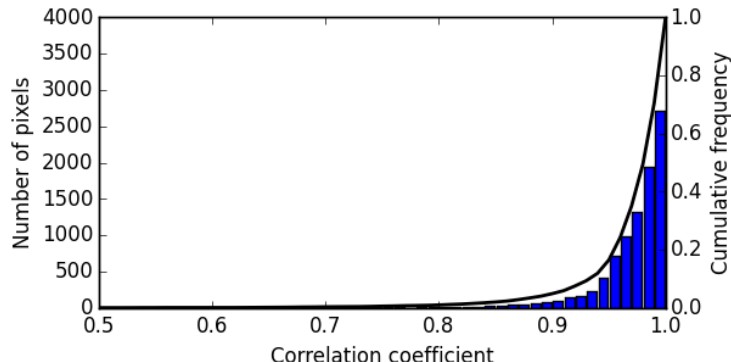

**Figure 7. Histogram and cumulative distribution of the correlation coefficient. Only 14 pixels out of 9,195 have a correlation coefficient less than 0.6.**

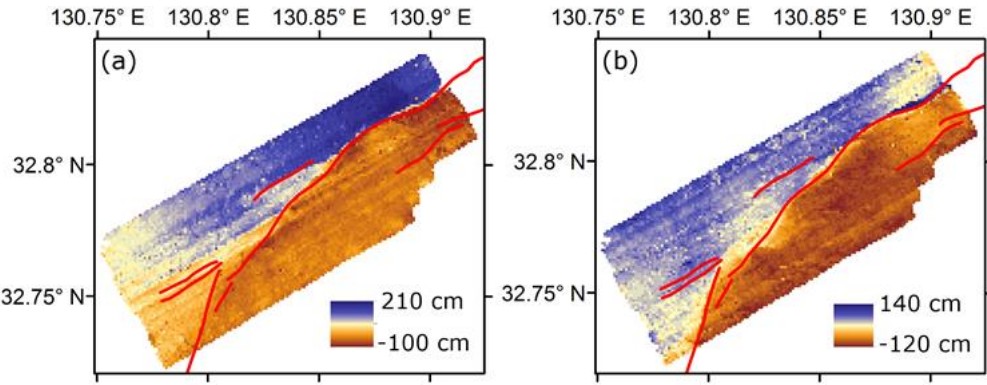

**Figure 8. East-west (a) and north-south (b) components of the coseismic displacement obtained from the maximum cross-correlation search of the Lidar DSMs.**

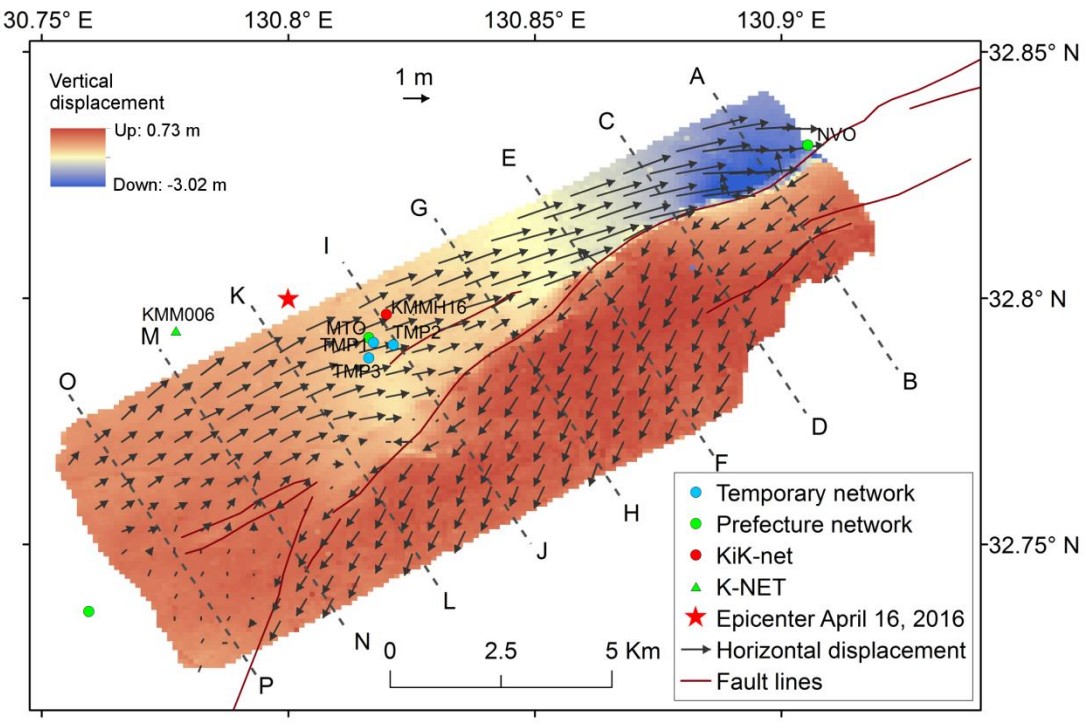

**Figure 9. Estimated three-dimentional coseismic displacement produced by the mainshock of the 2016 Kumamoto earthquake. The arrows indicate the amplitude and direction of the horizontal displacement at 500-m grid points.**

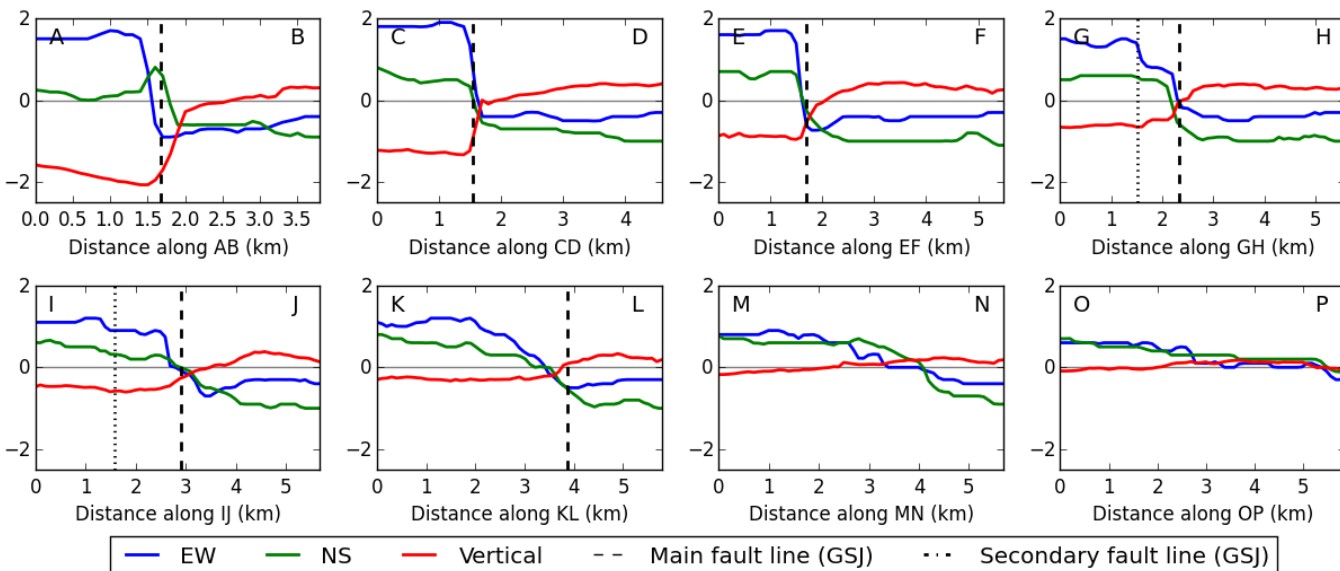

**Figure 10.** Estimated three-dimentional coseismic displacements estimated along the eight profile lines in Figure 9. Vertical break lines show the location of the known main Futagawa fault line by the GSJ. The location of the secondary fault line is indicated using dotted lines.

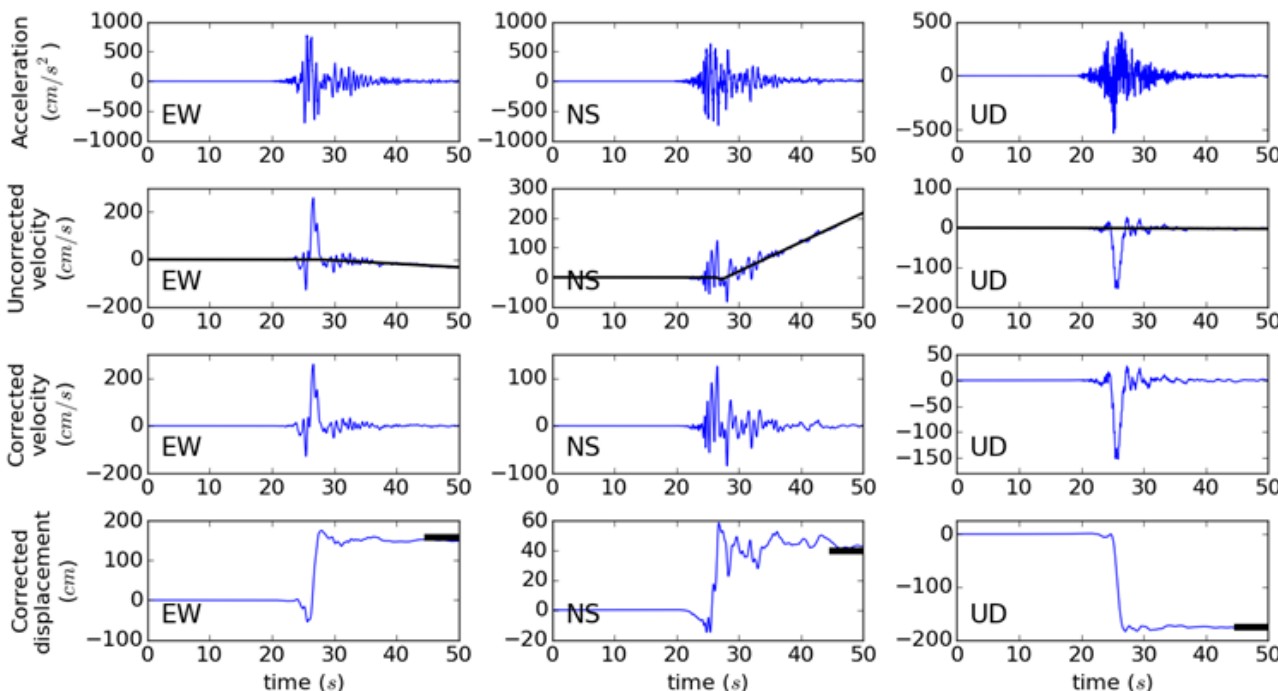

**Figure 11. Example of baseline correction procedure for the acceleration recorded at the Nishihara station. The trend of the uncorrected velocity was modeled by two straight lines based on the method by Wang et al. (2011) and was removed from the record. Then, the corrected displacement was calculated by integrating the acceleration with respect to time. The thick black line in the displacement time history represents the coseismic displacement calculated from the Lidar DSMs.**

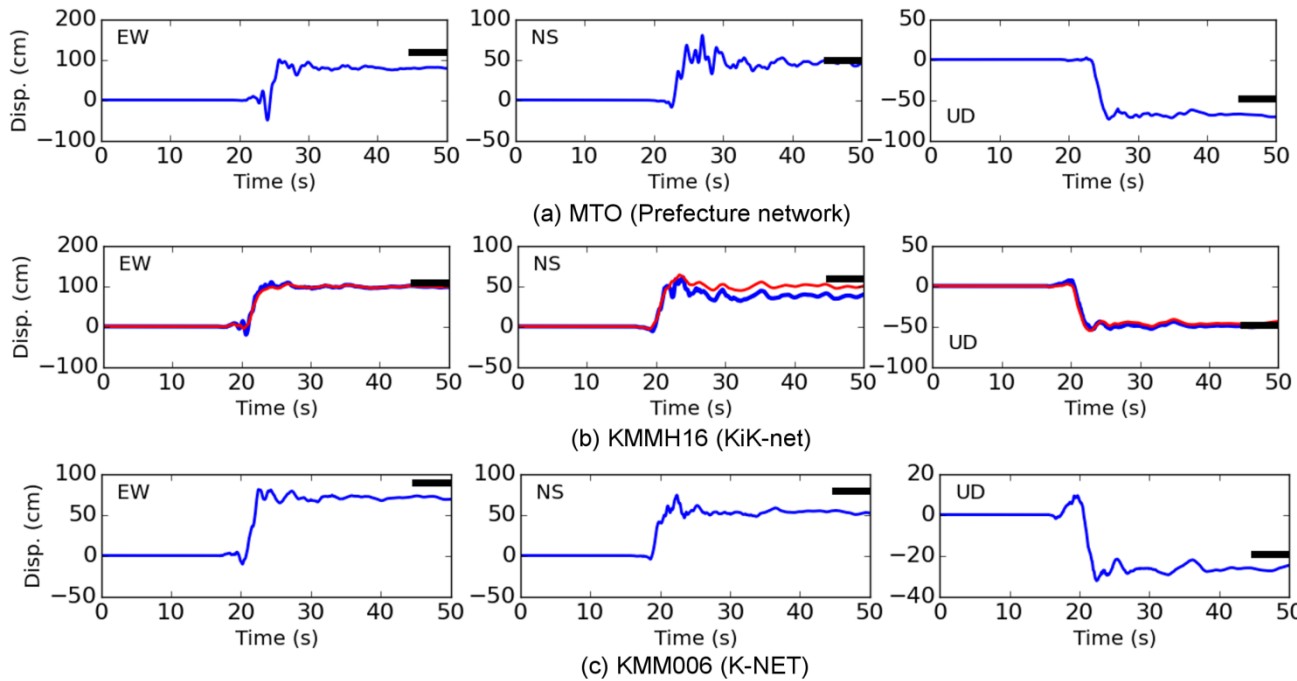

**Figure 12. Comparison of three-dimentional coseismic displacement obtained from Lidar DSMs (thick black line) and those obtained from the acceleration records at MTO station (a), KMMH16 KiK-net station (b), and KMM006 K-NET station (c). Red lines in KMMH16 KiK-net station show displacements at the bedrock (Ground level: -252 m). KMM006 K-NET station is located at 1 km from the nearest Lidar DSM point.**

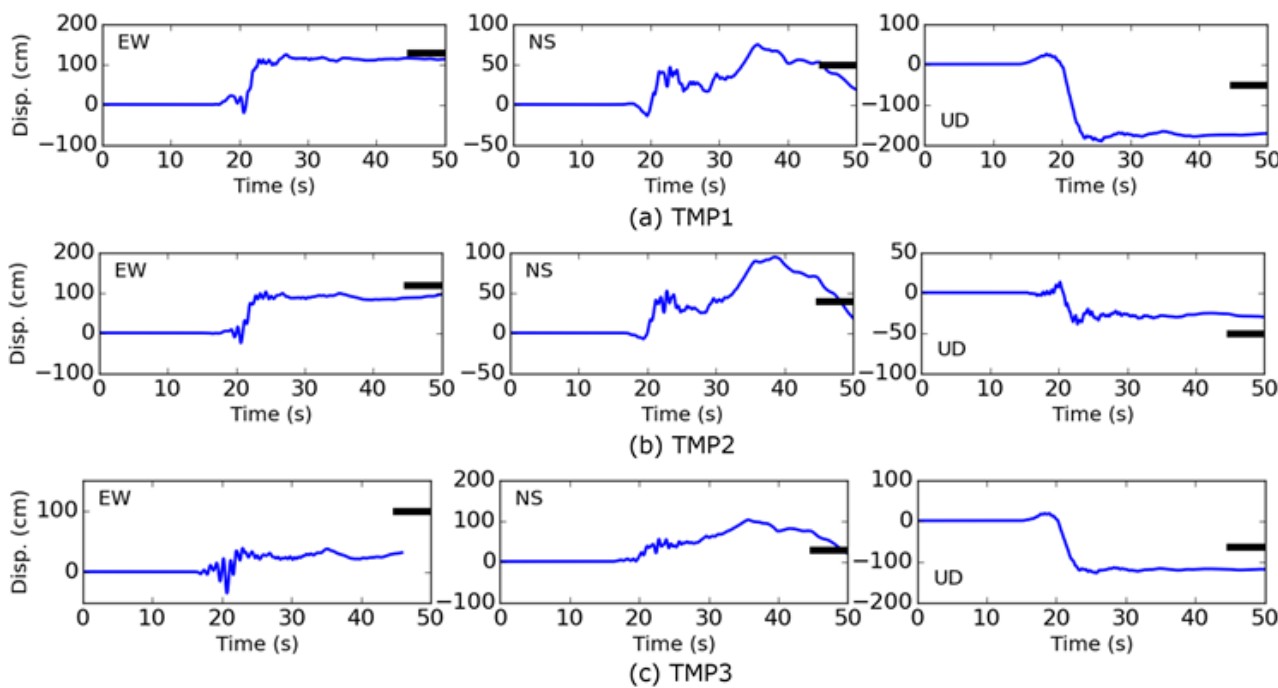

**Figure 13. Comparison of three-dimentional coseismic displacement obtained from Lidar DSMs (thick black line) with those obtained from the acceleration records at TMP1 (a), TMP2 (b), and TMP3 (c) stations.**

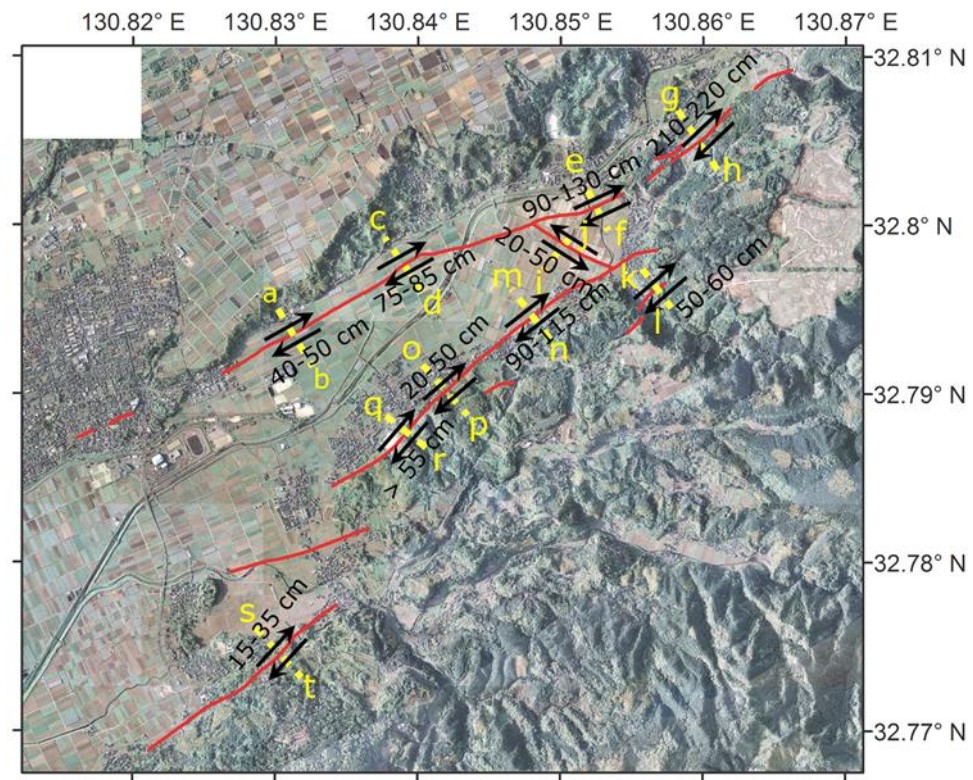

**Figure 14. Location of surface ruptures (red lines) observed during the field surveys of the GSJ (2016) and plotted on aerial images acquired by the Asia Air Co. on April 23. The black arrow represents the direction and amplitude of the observed strike slip at each location.**

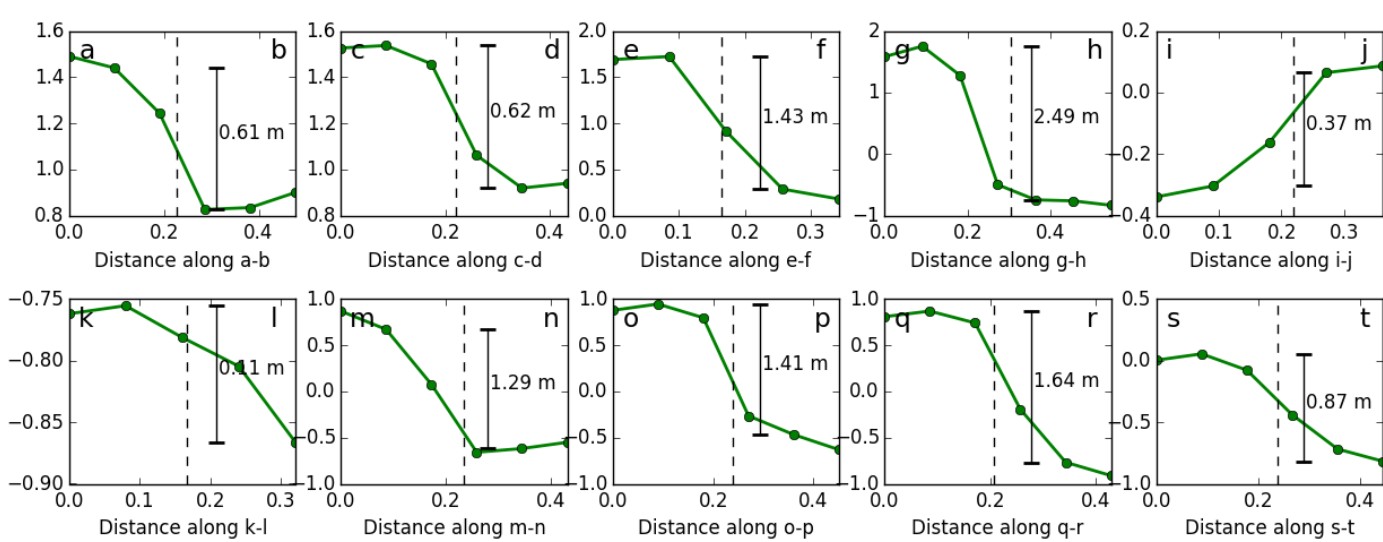

**Figure 15. Estimated coseismic displacement parallel to the fault lines along the ten profile lines including the locations of the field observation by the GSJ shown in Figure 14.**

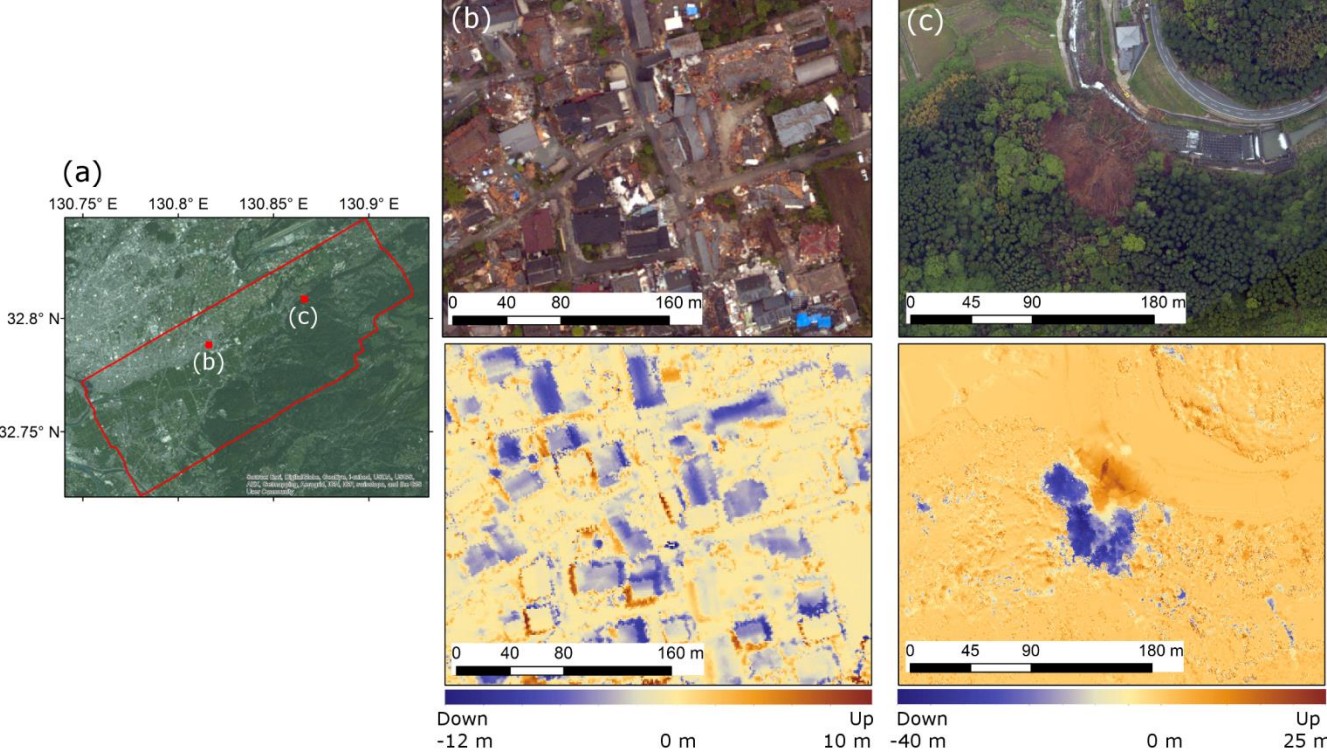

**Figure 16. Illustration of collapsed buildings and landslide along with the difference between the Lidar DSMs: location of the sample sites (a), a heavily damaged residential area (b), and forest including landslide (c). The top figures in (b) and (c) show aerial images taken on April 23 while the bottom figures show the differences between the two DSMs.**

