# Peer review of "Calculation of coseismic displacement from Lidar data in the 2016 Kumamoto, Japan, earthquake"

_Natural Hazards and Earth System Sciences, 2016_

## Referee Comment (RC1) · Anonymous Referee #1 · 17 Oct 2016

General comments:

The authors present a study that calculates the coseismic displacement of the recent Kumamoto earthquake from Lidar data and compares the results with the outcomes of alternative approaches based on strong-motion data. The manuscript is well written, scientifically sound and the topic is timely and certainly of interest for a wider group of readers in disaster risk management. I have, however, some comments and questions throughout the paper. Mainly, I suggest to improve the introduction, the discussion and conclusion chapters of the paper. For these reasons, I recommend the manuscript for publication in NHESS after minor revision.

Specific comments:

Introduction: I strongly suggest to add more references and a more in-depth review of

the state-of-the-art. In particular more emphasize should be given in presenting other studies that use Lidar and/or strong-motion data to estimate coseismic displacement. Based on this and the review of work that has been done related to this particular earthquake, it would be important to highlight the need for this study and the added-value that it can bring to the scientific understanding of the earthquake. I also suggest to remove the paragraph related to a general definition of Lidar which does not add much to the content.

Page 4, line 1: I suggest to move the references to Liu et al 2011 and Lie and Yamazaki 2013 to the introduction.

Page 4, line 23: The pixel resolution would be "increased" if you resampled from 50 cm to 10 cm. A few more words on the applied convolution method would be desirable.

Page 5, line 31: "Lidar data are capable. . ." I suggest to move this paragraph into a separate discussion section.

Page 7, line 17-24: Suggest to move this paragraph into a separate discussion section.

Discussion: Please add a separate discussion section that clearly outlines the limitations and benefits of the applied method, and compares the results with findings of other studies (linked to studies introduced in the introduction section).

Conclusions: Some more relevant conclusions would be desirable. For example, on the basis of your study, would you be able to say if it is worth to do the expensive Lidar surveys or would the other available sensor data have been enough to estimate the coseismic displacement to a sufficient degree? On the basis of this, what are the implications of your study on a better understanding of the earthquake physics? Is Lidar just a another data source to be used or does it make a difference with respect to the other data source that have been available for this event?

Technical corrections:

Page 1, Line 24: Suggest to rephrase sentence to: "This Kumamoto earthquake sequence triggered secondary effects such as landslides and liquefaction, and caused extensive damage to lifeline systems, buildings, bridges and transportation structures."

Page 4, line 32: ". . .vertical axis shows is used for the number . . ." remove "is used for".

Page 6, line 6: Add a reference to Fig. 1 to the statement that there are no Geonet stations in the area.

Page 6, line 19: Suggest to replace second "because" by "as a result of".

―――――――――――――――――――

---

## Referee Comment (RC2) · Anonymous Referee #2 · 30 Oct 2016

General comments: This manuscript presents a methodology to calculate co-seismic displacement using LIDAR data acquired after the recent earthquakes occurred in Kumamoto, Japan and the result were validated with ground motion records around. The proposed methodology represents a good alternative to monitor ground deformation using remote sensing data showing great potential to be used in disaster management and would be worthy of NHESS publication after minor revision.

I would like to suggest going deeper in the literature review. For instance, one point that was not mention is the advantages and/or disadvantages of the proposed methodology compared with other methods considering several aspects such as data availability, data coverage, application for disaster management assessment.

Specific comments: Page 2, line 17: Since this method calculates the permanent dis-

placement based on two different DSM scenes, it would be interesting to mention the time of acquisition of the LIDAR data.

Page 4, line 23: Please, explain why the original data was interpolated to 10 cm, and why not to 25 cm or 5 cm? Is any optimal spatial resolution considering factors such as detail of analysis, computational power?

Page 4, line 29: I understand why 201 x 201 pixel window was chosen, it however does not clearly explain why largest windows size can not be used too. Related with the previous comment, it would be interesting to see what is the relationship between the pixel size and the window size.

Page 6, line 26: Considering that a 201 x 201 pixel window is used to calculate the co-seismic deformation, please explain what is the selection criterion of the value from the DSM result that is compared with the displacement time history. On other words, is the value at the location of the seismic station used in this comparison?

Page 7, line 5: Although, the Figure 13 does show good agreement of the co-seismic displacement between the result using LIDAR data and the ground motion records, a quantitatively validation would be more convincing. For instance, the result of this methodology can be correlated with the result obtained from DInSAR analysis of PALSAR-2 data conducted by the Geospatial Information Authority of Japan (http://www.gsi.go.jp/BOUSAI/H27-kumamoto-earthquake-index.html)

Minor comments are printed in the attached document.

Please also note the supplement to this comment:
http://www.nat-hazards-earth-syst-sci-discuss.net/nhess-2016-315/nhess-2016-315-RC2-supplement.pdf

**Supplement:**

[revised manuscript text omitted]

---

## Author Comment (AC1) · 8 Nov 2016

Thank you very much for your insightful review. We consider your comments and suggestions in order to improve our manuscript. Details for each comment are addressed below.

General comments: The authors present a study that calculates the coseismic displacement of the recent Kumamoto earthquake from Lidar data and compares the results with the outcomes of alternative approaches based on strong-motion data. The manuscript is well written, scientifically sound and the topic is timely and certainly of interest for a wider group of readers in disaster risk management. I have, however, some comments and questions throughout the paper. Mainly, I suggest to improve the introduction, the discussion and conclusion chapters of the paper. For these reasons,

[Figure]

I recommend the manuscript for publication in NHESS after minor revision.

Specific comments:

COMMENT FROM REFEREE:

Introduction: I strongly suggest to add more references and a more in-depth review of the state-of-the-art. In particular more emphasize should be given in presenting other studies that use Lidar and/or strong-motion data to estimate coseismic displacement. Based on this and the review of work that has been done related to this particular earthquake, it would be important to highlight the need for this study and the added-value that it can bring to the scientific understanding of the earthquake. I also suggest to remove the paragraph related to a general definition of Lidar which does not add much to the content.

Author's response:

The referee is right to point out that more review of the state-of-art is necessary. We have extended the introduction by referring previous publications. We basically added publications related to the use of Lidar data to ground deformation, the use of only post-earthquake Lidar data, and the few cases where there was available pre- and post- earthquake data. We also emphasize the additional information that could be acquired from Lidar data. About to the general definition of Lidar, since NHESS journal gathers readers from various disciplines, we believe that the definition of Lidar would be useful to an uninitiated reader

Author's changes in manuscript:

The extended information in the introduction was added from page 2, line 10 to page 3, line 32, as follows: "
[revised manuscript text omitted]

COMMENT FROM REFEREE:

Page 4, line 1: I suggest to move the references to Liu et al 2011 and Lie and Yamazaki 2013 to the introduction.

Author's response:

In accordance with the referee's comment, we have moved the references to the introduction.

Author's changes in manuscript:

The references are located between page 2, line 31 to page 3, line 1, as follows: "...Liu et al. (2011) extracted the shifts of vehicles between the panchromatic and multispec-

tral QuickBird images, which were taken with a time lag of approximately 0.2 seconds, and then they estimated the vehicles' velocity. Liu and Yamazaki (2013) calculated the crustal displacement during the 2011 Mw 9.0 Tohoku earthquake by estimating the shift of undamaged buildings using the cross–correlation coefficient between the TerraSAR–X intensity images taken before and after the earthquake. . .."

COMMENT FROM REFEREE:

Page 4, line 23: The pixel resolution would be "increased" if you resampled from 50 cm to 10 cm. A few more words on the applied convolution method would be desirable.

Author's response:

The referee is right to point out this mistyping. We apologize for it.

Author's changes in manuscript:

In page 6, line 8-11, as follows: ". . .The pixel resolution was increased from 50 cm to 10 cm by using the cubic convolution method, where a bicubic function is fitted using 4x4 pixels neighborhood and used to estimate the intermediate values. The subpixel size was decided based on the computational effort that is required to detect the peak value of the correlation coefficient. . ."

COMMENT FROM REFEREE:

Page 5, line 31: "Lidar data are capable. . ." I suggest to move this paragraph into a separate discussion section.

Author's response:

In accordance with the referee's comment, we have moved this paragraph to a new section.

Author's changes in manuscript:

The paragraph has been moved to page 9, lines 13-17.

[Figure]

COMMENT FROM REFEREE:

Page 7, line 17-24: Suggest to move this paragraph into a separate discussion section.

Author's response:

In accordance with the referee's comment, we have moved this paragraph to a new section.

Author's changes in manuscript:

The paragraph has been moved to page 9, lines 7-12.

COMMENT FROM REFEREE:

Discussion: Please add a separate discussion section that clearly outlines the limitations and benefits of the applied method, and compares the results with findings of other studies (linked to studies introduced in the introduction section).

Author's response:

In accordance with the referee's comment, we added a new section where we addressed the suggestions.

Author's changes in manuscript:

The new section is located at page 8, line 26: "6 Discussion Our result could recover the spatial distribution of the three-dimensional (east-west, north-south, and up-down) coseimsic displacement and validated the fault line drawn by the GSJ (Error! Reference source not found., 8 and 9). From the evaluation of the parameters used, the results were found to be highly sensitive to the window size. Basically, it is crucial that the windows have to cover several features, such as buildings, trees and different topography, in order to obtain a clear peak value in the correlation coefficient (Error! Reference source not found.c). This issue was our main concern in agricultural fields because large areas have uniform elevation. In this study, a constant window size

was used; however, if the land use information is available, different window sizes can be applied. For instance, in urban areas the window size can be smaller than that for agricultural lands. Therefore, one limitation of the method is the required window size because the larger the window size, the lower the spatial resolution of coseismic displacement. Comparing our result with the InSAR satellite images published by the GSI, our result provides the 3D coseismic displacement; while the InSAR results provide only the displacement to the line-of-sight. But concerning about the area coverage, satellite sensors can cover a larger area than airborne Lidar sensors do. The slips calculated from our results are very close to that obtained from the field observation for most cases (Error! Reference source not found. and 15). It is observed that in the majority of the cases our results are greater than the measured ones. We believe that the main reason for this is that the type of soil is cohesive in this area. Cohesive soils have the ability to exhibit large plastic deformation that depends on the water content and, as can be seen, the area is mostly used for agricultural purposes where the soil has high water content. Thus, the surface rupture measured in the field might not be the total slip. The largest differences between the GSJ survey and the Lidar results are observed in the profiles 'op' and 'qr'. Lidar data are capable of extracting other types of information. Error! Reference source not found. Error! Reference source not found.shows two areas: one with collapsed buildings and the other where a landslide occurred. Error! Reference source not found. also shows the change in elevations between the DSMs after removing the horizontal coseismic displacement. As can be observed, the large change in elevations implies that a building collapsed or a landslide occurred. Therefore, with proper thresholds, these phenomena can be detected automatically. This issue will be discussed in a future publication."

COMMENT FROM REFEREE:

Conclusions: Some more relevant conclusions would be desirable. For example, on the basis of your study, would you be able to say if it is worth to do the expensive Lidar surveys or would the other available sensor data have been enough to estimate

the coseismic displacement to a sufficient degree? On the basis of this, what are the implications of your study on a better understanding of the earthquake physics? Is Lidar just a another data source to be used or does it make a difference with respect to the other data source that have been available for this event?

Author's response:

In accordance with the referee's comment, we extended the conclusion section where we addressed the suggestions.

Author's changes in manuscript:

The additional information of the conclusions section is located at page 9, line 30: "...The detailed information of coseismic displacement is indeed useful to constrain the focal mechanism of the event. Recall that the GSI's preliminary report estimated a slip of about 24 m in the source zone during the 2011 Mw 9.0 Tohoku earthquake from an inversion method using the inland GEONET station records. However, later Sato et al. (2011) observed a coseismic displacement of 23 m at the ocean bottom and pointed out that this information could better constrain the focal mechanism. Thus, our results, which records higher coseismic displacement than the one recorded from GNSS stations, would improve the source estimation. However this issue is out of the scope of this paper and will be addressed in a future publication. As mentioned before, there are only few cases in which Lidar data before and after an earthquake are available. The main reason is a high cost of Lidar surveys. However, this technology can be used properly for a specific region of interest, such as along fault lines. For instance, the B4 project (Bevis et al., 2005) collected Lidar data of the southern San Andreas and San Jacinto faults in southern California in order to have a pre-event Lidar data for future earthquakes"

Technical corrections:

COMMENT FROM REFEREE:

Page 1, Line 24: Suggest to rephrase sentence to: "This Kumamoto earthquake sequence triggered secondary effects such as landslides and liquefaction, and caused extensive damage to lifeline systems, buildings, bridges and transportation structures."

Author's response:

In accordance with the referee's comment, we have changed the statement.

Author's changes in manuscript:

Page 1, line 24

COMMENT FROM REFEREE:

Page 4, line 32: "...vertical axis shows is used for the number..." remove "is used for".

Author's response:

The referee is right to point out this mistyping. We apologize for it.

Author's changes in manuscript:

The words "is used for" was removed from the sentence. Page 6, line 19 "...vertical axis shows the number ..."

COMMENT FROM REFEREE:

Page 6, line 6: Add a reference to Fig. 1 to the statement that there are no Geonet stations in the area.

Author's response:

Adding the reference make the sentence clearer. We appreciate the referee for this suggestion.

Author's changes in manuscript:

Page 7, line 20: "Unfortunately, there is no GEONET station in this study area (Error!

Reference source not found.). . ."

COMMENT FROM REFEREE:

Page 6, line 19: Suggest to replace second "because" by "as a result of".

Author's response:

In accordance with the referee's comment, we have changed the statement. Please note that this paragraph have been moved to the introduction section, where we added more detail about displacement from acceleration records.

Author's changes in manuscript:

Page 3, line 12: ". . .because the baseline is shifted as a result of several factors. . ."

Please also note the supplement to this comment:
http://www.nat-hazards-earth-syst-sci-discuss.net/nhess-2016-315/nhess-2016-315-AC1-supplement.pdf

―――――――――――――――――

---

## Author Comment (AC2) · 8 Nov 2016

Thank you very much for your insightful review. We consider your comments in order to improve our manuscript. Details for each one are addressed below.

General comments: This manuscript presents a methodology to calculate co-seismic displacement using LIDAR data acquired after the recent earthquakes occurred in Kumamoto, Japan and the result were validated with ground motion records around. The proposed methodology represents a good alternative to monitor ground deformation using remote sensing data showing great potential to be used in disaster management and would be worthy of NHESS publication after minor revision.

COMMENT FROM REFEREE:

[Figure]

I would like to suggest going deeper in the literature review. For instance, one point that was not mention is the advantages and/or disadvantages of the proposed methodology compared with other methods considering several aspects such as data availability, data coverage, application for disaster management assessment

Author's response:

As recommended by the referee, we have increased the introduction, where we address his suggestions.

Author's changes in manuscript:

The extended information in the introduction was added from page 2, line 10 to page 3, line 32, as follows: " 
[revised manuscript text omitted]

Specific comments:

COMMENT FROM REFEREE:

Page 2, line 17: Since this method calculates the permanent displacement based on two different DSM scenes, it would be interesting to mention the time of acquisition of the LIDAR data.

Author's response:

In accordance with the referee's comment, we added this information.

Author's changes in manuscript:

Page 4, line 1

COMMENT FROM REFEREE:

Page 4, line 23: Please, explain why the original data was interpolated to 10 cm, and why not to 25 cm or 5 cm? Is any optimal spatial resolution considering factors such as detail of analysis, computational power?

Author's response:

As pointed out by the referee, the main reason was the computational power.

Author's changes in manuscript:

The sentence was modified and is located in page 6, line 8: "...The pixel resolution was increased from 50 cm to 10 cm by using the cubic convolution method, where a bicubic function is fitted using 4x4 pixels neighborhood and used to estimate the intermediate values. The subpixel size was decided based on the computational effort that is required to detect the peak value of the correlation coefficient...."

COMMENT FROM REFEREE:

Page 4, line 29: I understand why 201 x 201 pixel window was chosen, it however does not clearly explain why largest windows size can not be used too. Related with the previous comment, it would be interesting to see what is the relationship between the pixel size and the window size.

Author's response:

Indeed the use of larger window-size is possible, however, it will reduce the resolution of the coseismic displacement. We address this in page 5, line 28: "However, there exists a trade-off between the size of the window and resolution because the resolution of the spatial variation of the coseismic displacement decreases with the increase in the size of the window." With respect of the relationship of the pixel size and the window size. We believe that what is more important is the amount of different features that can be covered in the window size. Because it will help to detect the peak value of the correlation coefficient. For sure, the resolution of the pixel is relevant to define the features; but, we believe that a pixel size of 50 x 50 cm is enough to clearly define features such as buildings, trees, and changes in topography. This issue is stated in newly added Discussion section in Page 8.

Author's changes in manuscript:

Page 8, line 28: "...From the evaluation of the parameters used, the results were found to be highly sensitive to the window size. Basically, it is crucial that the windows have to cover several features, such as buildings, trees and different topography, in order to obtain a clear peak value in the correlation coefficient (Error! Reference source not found.c). This issue was our main concern in agricultural fields because large areas have uniform elevation. In this study, a constant window size was used; however, if the land use information is available, different window sizes can be applied. For instance, in urban areas the window size can be smaller than that for agricultural lands..."

COMMENT FROM REFEREE:

Page 6, line 26: Considering that a 201 x 201 pixel window is used to calculate the co-seismic deformation, please explain what is the selection criterion of the value from the DSM result that is compared with the displacement time history. On other words, is the value at the location of the seismic station used in this comparison?

Author's response:

Yes, the coseismic displacement from Lidar data used in the comparison comes from the window that contains the strong-motion station.

Author's changes in manuscript:

We modified the sentence to clarify this question. We appreciate the referee for this suggestion. Page 8, line 3: "...The coseismic displacement calculated from the Lidar data at the same location of the strong-motion station, shown as a black thick line..."

COMMENT FROM REFEREE:

Page 7, line 5: Although, the Figure 13 does show good agreement of the co-seismic displacement between the result using LIDAR data and the ground motion records, a quantitatively validation would be more convincing. For instance, the result of this methodology can be correlated with the result obtained from DInSAR analysis of PALSAR-2 data conducted by the Geospatial Information Authority of Japan (http://www.gsi.go.jp/BOUSAI/H27-kumamoto-earthquake-index.html).

Author's response:

The comparison of our results from the Lidar data with the coseismic displacements from strong-motion records is quantitative. Besides, we consider that the comparison with the displacement measured at the surface rupture points in the field surveys is conclusive considering that this is tangible information. We showed the comparison with the figures for better understanding. With all due respect we believe that a comparison of our results with SAR data will not contribute so much to our manuscript. As stated in Introduction, DInSAR results are to the line-of-sight (LOS), not the 3D displacement.

Author's changes in manuscript:

No changes.

Minor changes

COMMENT FROM REFEREE:

Page 1, line 29: Change "...Japan Aerospace..." to "... The Japan Aerospace..."

Author's response:

In accordance with the referee's comment, we have change the sentence.

Author's changes in manuscript:

The change is located at page 1, line 29.

COMMENT FROM REFEREE:

Page 1, line 30: Change "...sensor PALSAR-2..." to "...PALSAR-2 sensor..."

Author's response:

In accordance with the referee's comment, we have change the sentence.

Author's changes in manuscript:

The change is located at page 1, line 30.

COMMENT FROM REFEREE:

Page 2, line 4: Change "...the authors of this paper..." to "...we..."

Author's response:

With all due respect to the referee, we prefer to keep the original version.

Author's changes in manuscript:

No changes.

COMMENT FROM REFEREE:

Page 2, line 4: Change "...calculated the spatial coherence values (International Charter, 2016), which could highlight the extensive landslides and severe damages to buildings 5 along the Futagawa fault line. ..." to "...calculated the coherence image that shows the extensive landslides and severe damages to buildings 5 along the Futagawa fault line (International Charter, 2016). ..."

Author's response:

With all due respect to the referee, we prefer to keep the original version.

Author's changes in manuscript:

No changes.

COMMENT FROM REFEREE:

Page 2, line 6; Change "...earthquake, the government agencies..." to "...earthquake, government agencies..."

Author's response:

In accordance with the referee's comment, we have change the sentence.

Author's changes in manuscript:

The change is located in the new version of the manuscript at page 2, line 7.

COMMENT FROM REFEREE:

Page 2, line 7; Change "...As well as..." to "...such as..."

Author's response:

In accordance with the referee's comment, we have change the sentence.

Author's changes in manuscript:

The change is located in the new version of the manuscript at page 2, line 8.

Comment from referee:

Page 2, line 8: Delete ". . .were also conducted. . ."

Author's response:

In accordance with the referee's comment, we have change the sentence.

Author's changes in manuscript:

The change is located in the new version of the manuscript at page 2, line 9.

COMMENT FROM REFEREE:

Page 2, line 17: Change ". . .one just after. . ." to ". . .one soon after. . ."

Author's response:

In accordance with the referee's comment, we have change the sentence.

Author's changes in manuscript:

The change is located in the new version of the manuscript at page 4, line 1.

COMMENT FROM REFEREE:

Page 2, line 18: Change ". . .is available. . ." to ". . .were used. . ."

Author's response:

In accordance with the referee's comment, we have change the sentence.

Author's changes in manuscript:

The change is located in the new version of the manuscript at page 4, line 2.

COMMENT FROM REFEREE:

Page 2, line 22: Change ". . .a day after. . ." to ". . .one day after. . ."

Author's response:

In accordance with the referee's comment, we have change the sentence.

Author's changes in manuscript:

The change is located in the new version of the manuscript at page 4, line 7.

COMMENT FROM REFEREE:

Page 2, line 24: Change ". . .Furthermore, because of an unexpected. . ." to ". . .Due to the. . ."

Author's response:

With all due respect to the referee, we prefer to keep the original version.

Author's changes in manuscript:

No changes.

COMMENT FROM REFEREE:

Page 2, line 25: Change ". . .acquire the Lidar data. . ." to ". . .acquire Lidar data. . ."

Author's response:

In accordance with the referee's comment, we have change the sentence.

Author's changes in manuscript:

The change is located in the new version of the manuscript at page 4, line 10.

COMMENT FROM REFEREE:

Page 2, line 28: Change "... For the sake of brevity,..." to "...Here,..."

Author's response:

With all due respect to the referee, we prefer to keep the original version.

Author's changes in manuscript:

No changes.

COMMENT FROM REFEREE:

Page 2, line 31: Change "...most parts of the town of Mashiki and a few parts of the town of Kashima, the town of Mifune,..." to "...most parts of the Mashiki town and a few parts of Kashima town, Mifune town..."

Author's response:

In accordance with the referee's comment, we have change the sentence.

Author's changes in manuscript:

The change is located in the new version of the manuscript at page 4, line 16.

COMMENT FROM REFEREE:

Page 3, line 15: Change "...more clear evidence..." to "...more clearly evidence..."

Author's response:

Considering the referee's comment, we have change the sentence to "...a clearer evidence..."

Author's changes in manuscript:

The change is located in the new version of the manuscript at page 4, line 32.

COMMENT FROM REFEREE:

Page 3, line 15: Change "...an overlap..." to "...color composite image..."

Author's response:

The referee is right to point out that the technical word is color composite image. However, since NHESS gathers readers from different disciplines, we prefer to show the definition rather than the technical term..

Author's changes in manuscript:

No changes.

COMMENT FROM REFEREE:

Page 5, line 7: Change "...occurred because of the mainshock..." to "...occurred as a result of the mainshock..."

Author's response:

In accordance with the referee's comment, we have change the sentence.

Author's changes in manuscript:

The change is located in the new version of the manuscript at page 6, line 25.

COMMENT FROM REFEREE:

Page 13, line 2: Please add north direction at Figure 4

Author's response:

In accordance with the referee's comment, we have modified the Figure 4.

Author's changes in manuscript:

The change is located in the new version of the manuscript at page 16, line 1.

Please also note the supplement to this comment:

http://www.nat-hazards-earth-syst-sci-discuss.net/nhess-2016-315/nhess-2016-315-AC2-supplement.pdf

[Figure]

**Supplement:**

[revised manuscript text omitted]